# *Listeria* Occurrence in Conventional and Alternative Egg Production Systems

**DOI:** 10.3390/microorganisms11092164

**Published:** 2023-08-27

**Authors:** Steven C. Ricke, Corliss A. O’Bryan, Michael J. Rothrock

**Affiliations:** 1Meat Science and Animal Biologics Discovery Program, Department of Animal and Dairy Sciences, University of Wisconsin, Madison, WI 53706, USA; 2Food Science Department, University of Arkansas, Fayetteville, AR 72704, USA; cobryan@uark.edu; 3U.S. National Poultry Research Center, Egg Safety & Quality Research Unit, USDA-ARS, Athens, GA 30606, USA; michael.rothrock@usda.gov

**Keywords:** *Listeria*, poultry, eggs, layer housing environments

## Abstract

*Listeria* continues to be a persistent foodborne pathogen that is responsible for human cases of listeriosis when contaminated food products are consumed. Human subjects considered to be most susceptible include the elderly, immunocompromised, and pregnant women. *Listeria* is characterized as a saprophytic organism with the capability of responding and adapting to constantly changing environments because it possesses multiple stress response mechanisms to overcome varying temperatures, salt concentrations, and pH, among others. Primary foods and food products associated with listeriosis include dairy products and ready-to-eat meats such as turkey products. Historically, chicken eggs have not been identified as a primary source of *Listeria,* but the potential for contamination during egg production and processing does exist. *Listeria* species have been isolated from egg-processing plant equipment and are presumed to occur in egg-processing plant environments. Whether *Listeria* is consistently disseminated onto eggs beyond the egg-processing plant is a risk factor that remains to be determined. However, research has been conducted over the years to develop egg wash solutions that generate combinations of pH and other properties that would be considered inhibitory to *Listeria*. Even less is known regarding the association of *Listeria* with alternative egg production systems, but *Listeria* has been isolated from pasture flock broilers, so it is conceivable, given the nature of the outdoor environments, that layer birds under these conditions would also be exposed to *Listeria* and their eggs become contaminated. This review focuses on the possibility of *Listeria* occurring in conventional and alternative egg-laying production and processing systems.

## 1. Introduction

*Listeria monocytogenes* is a foodborne pathogen of notable concern, especially due to its growth under household refrigeration temperatures (40 °F, 4 °C) and at a pH range of 4.5 to 9.6 [1,2,3]. *Listeria monocytogenes* has been subdivided into four phylogenetic evolutionary lineages due to variations in ecology recombination rates and genomic content [4,5]. Lineage I consist of most human clinical isolates and includes serotypes 1/2b, 3b, 3c, and 4b strain [5,6]. Lineage II is more prominent in environmental samples, foods, and animals and includes 1/2a, 1/2c, and 3a, while Lineages III and IV are mainly found in ruminants [7]. In human cases of listeriosis stemming from chicken meat, serotype 1/2b is the most prominent [8]. However, serotypes 1/2a and 1/2b are predominately isolated from ready-to-eat (RTE) foods [9,10], and Gilbreth et al. [11] noted that clinical isolates from patients made ill by RTE foods were mostly 1/2a or 4b. Severe infections of *L. monocytogenes* can occur in the elderly, immunocompromised, and pregnant women and can have a mortality rate exceeding 20% [12,13,14]. In 2021, the last year for which data are available, 2268 confirmed cases of listeriosis were reported by 30 EU/EEA Member States, a rate of 0.51 per 100,000 population [15]. In that same year in the U.S., there were approximately 1029 laboratory-confirmed domestically acquired cases, a rate of 0.31 per 100,000 [16]. The infective dose of *L. monocytogenes* varies with the strain of *L. monocytogenes* and the susceptibility of the victim, with pregnant women, newborns, the elderly, and immunocompromised persons being the most vulnerable [17]. Epidemiological modeling of the 2015 United States (U.S.) *Listeria* outbreak in ice cream indicated that in each gram of product, 620 colony forming units (CFU) *L. monocytogenes* cells were present and that the highly exposed population ingested 7.2 × 10^6^ to 3.3 × 10^7^ cells despite only four cases being reported [18]. Due to growth conditions and mortality rate, there is a “zero tolerance” policy for *Listeria* (1 CFU/25 g product) in ready-to-eat (RTE) foods in the U.S. [19].

While not as well documented as other foodborne pathogens, *Listeria* can exist on eggshells and within the egg [20,21]. However, it has not been established whether *Listeria* is a consistent foodborne pathogen associated with commercial egg production. Also, with the growing popularity of pasture flock layer hen operations, it remains unknown if hens maintained under these environmental conditions are more vulnerable to *Listeria* exposure. To the best of our knowledge, there has not been a review directly focusing on the potential for *L. monocytogenes* contamination in eggs. As such, this review investigates the relevant literature concerning *L. monocytogenes* contamination during egg production, including alternative systems such as pastured layer flock poultry, to ascertain potential risks.

The per capita consumption of eggs in the U.S. was 320 in 2022, while the Chinese consume the most eggs per capita at 400 [22]. There were 1627 trillion eggs produced globally in 2022, down from 1633 trillion in 2021, primarily due to a worldwide avian influenza outbreak [23]. Other eggs, such as duck, turkey, goose, quail, and ostrich eggs, may be sold; however, sales of these products are not reported to the USDA. Avian influenza has decimated many commercial egg producers, while organic, cage-free, and pasture-raised egg producers have not been as badly impacted [24]. This has, in turn, led to the specialty eggs being lower in price than conventional eggs and is driving the sales of these eggs higher [24].

As of December 2022, cage-free layers constituted 28.2% of all egg-laying hens, and predictions state that in order to meet demand, the percentage of cage-free layers will need to be 66% of all laying hens by 2026 [25]. This consumer interest has resulted in companies such as Walmart announcing plans to go cage-free; Walmart reported that it had converted 21% of eggs sold at Walmart stores and 41% of the eggs sold at Sam’s to cage-free as of 2023 [26]. Non-cage housing systems have been reported to increase farm-level costs by 40 to 70% due to an increase in feed costs, housing, and labor [27,28]. While pastured poultry eggs fit the criterion of non-caged housing, other management strategies include floor-raised, also known as pen-raised or deep litter-raised layer hens [29]. Pen-raised layer hens are raised in an environment similar to broilers and are provided 1.5 ft^2^ (0.14 m^2^) per hen, and pullets (less than a year old) are usually isolated from mature egg-laying hens to prevent trampling [29,30].

## 2. Egg Production and Food Safety: General Concepts

Raw eggs are not considered RTE foods and, in many cases, are cooked thoroughly, which inactivates pathogens [31]. However, the consumption of raw cookie dough or mayonnaise (which contains eggs), as well as raw egg use in protein shakes, suggests that they could serve as a potential vector for listeriosis [32]. Federal regulatory agencies in the U.S. do not oversee farms that contain less than 3000 laying hens, and these farms are not required to register under the USDA’s Egg Products Inspection Act (EPIA), which many pasture poultry farms may fall under [31,33,34].

The cartons of eggs consumers purchase have code dates printed on them that may be best by, best if used by, or an expiration date [35]. An expiration date can be no longer than 30 days from the date the eggs were packed into the carton, while sell-by or use-by dates can be no more than 45 days from the date the eggs were placed in the carton [35]. The USDA recommends that eggs be cooked to 160 F (71 °C) and that shell eggs should not be washed as bacteria such as *Salmonella* can be “sucked” into the egg through pores during washing [35]. USDA also recommends using pasteurized eggs or egg products when preparing recipes that call for using eggs raw or undercooked [35].

## 3. *Listeria* in Poultry Production

There are several studies regarding the contamination of broiler flocks and broiler processing plants with *L. monocytogenes.* These studies were extensively reviewed by Rothrock et al. [5,36] but are briefly discussed here. Cox et al. [37] found that 6% of eggshell fragments in a U.S. commercial broiler hatchery tested positive for *L. monocytogenes*. Despite this, some studies have reported a 5% and 14% prevalence of *Listeria* in fecal samples [38,39]. In the first round of rearing, before broilers were placed, pastured poultry grass and soil samples yielded 8 isolates of *Listeria,* while after broilers were added, 13 serotypes were isolated [40]. In the second round of rearing, no isolates were obtained from soil and grass before the introduction of broilers, but 19 isolates were found after broilers were placed [40]. In processing facilities, floor drains are an important niche for *Listeria* persistence. Blackman and Frank [41] and Berrang et al. [42] performed experiments to determine whether a spray of water into a contaminated drain could carry *Listeria* cells to broiler meat on a table 2.4 m away from the drain. In Brazil, 14.4% of samples collected from post-evisceration poultry products in a plant with automatic evisceration tested positive for *L. monocytogenes,* as opposed to a plant with manual evisceration, which had 19.4% positive [43]. On retail products, Miranda et al. [44] found that 49.1% of organic and 41% of conventionally grown broiler drumsticks were positive for *L. monocytogenes*. Elmali et al. [45] collected and cultured broiler wing meat purchased from supermarkets and determined that 57/120 samples were positive for *Listeria* spp. (47.5%), and 54 of the 57 samples were identified to be *L. monocytogenes*.

Aury et al. [46] also performed a risk assessment for broiler flocks before processing. They found that moisture in the litter, determined by the type of litter used as well as poorly cleaned drinking troughs, increased the presence of *Listeria.* This is likely due to the ability of *Listeria* to persist in humid organic material and form biofilms [46,47,48]. A dirty and clean area in the poultry house and pest management were also required to reduce *Listeria* contamination along with reducing other foodborne pathogens [46]. They also found that workers were considered mechanical vectors of contamination from the environment, and hygiene measures such as food dips, hand washing, and changing clothes and shoes were instrumental in reducing *Listeria* contamination and flock-to-flock cross-contamination [46].

As Rothrock et al. [5,36] indicated, these data demonstrate the potential for broilers to be contaminated with *Listeria* and transmit that pathogen into the food supply while persisting in processing environments and environmental samples. Therefore, these data suggest that egg-laying production may also be subject to *Listeria* contamination and add potential evidence beyond the limited data available for *Listeria* in layer hen production and egg processing. However, it is important to note that broiler and layer operations vary considerably, and these must be considered when evaluating pathogen contamination and broiler and layer production.

## 4. Layer Hens and Egg Production Systems

Commercial layer hens begin their life in the hatchery after a 21-day incubation and are placed on a farm within 12 to 48 h of hatching [49]. In cage-based production systems, these immature chickens, commonly referred to as pullets, are raised in cages separate from the mature birds for 18 weeks and are not given more than 10 h of daily light to prevent premature egg laying [49]. Birds are subsequently transferred to a layer house and placed in larger cages; they begin producing eggs around 20 to 21 weeks and efficiently lay for 52 to 60 weeks [50]. The birds then typically begin to go through a molting process to stop egg production, followed by rejuvenation for another egg-laying cycle [51]. After molting, layer hens usually produce larger but poorer-quality eggs for a shorter period [49]. Depending on economic factors, including flock size and feed costs, hens may undergo one to three laying cycles before they are replaced [49].

Farmers, particularly those of backyard flocks, may choose to hatch their eggs instead of receiving them from a hatchery. However, small chicks are at risk from predation from rats, as well as household pets such as dogs and cats [52]. The USDA defines free-range as any system that provides limited access to a fenced-in outdoor area [53,54]. Pasture poultry refers to using a mobile coop with nest boxes with constant daytime access to the outdoors and typical coop rotation to ensure pasture vitality [53,54]. However, this definition varies as the certified humane organization requires 2 ft^2^ (0.2 m^2^) of outdoor access to the outdoors and at least six hours of access to the outdoors to be considered free-range, and for pasture-raised birds, 108 ft^2^ (10 m^2^) per bird of grass is required with the fields being rotated [55]. The University of Maryland Extension office defined free-range as having movable housing with access to pasture and pasture-poultry as using a field pen to control their grazing area and given a continued supply of forage [52]. Due to the variation in the definition, it can, therefore, be difficult to assess the impact these systems have on egg-laying production.

Ferrante et al. [56] compared egg production in organic versus barn-raised layers at the peak of laying and determined that egg laying was slightly higher in organic hens (94.5%) than in barn hens (93.0%). Birds raised on deep litter, as opposed to those raised in cages, had reduced egg production, reduced egg weight, and higher mortality [57]. Castellini et al. [58] found that organically raised birds exhibited a lower egg deposition level (64.1%) as compared to the control (73.9%) and suggested this was due to a need for more intense motor activity and lower protein and energy uptake due to the consumption of grass. Zaheer [22] noted that while a typical bird produces 250 to 300 eggs/year, this drops considerably to 180 to 200/year when in hot and humid conditions. As such, free-range and pasture-poultry systems may inadvertently impact egg laying by exposing the hens to inclement weather and seasonal temperature variations.

## 5. *Listeria* Presence in Layer Hens and the Environment

Free-range and pasture systems can expose the hens to wild birds, rodents, and other vectors that can harbor *Listeria* [36,59,60]. As in humans, *L. monocytogenes* can cause septicemia and localized encephalitis in adult poultry; young chicks are more susceptible to a chronic form of infection, and the mortality rate in these young birds can be as high as 40% [61]. Adult chickens and turkeys are relatively resistant to experimental infection and are thought to be prime reservoirs for contamination of the litter and environment of poultry production houses [62,63]. The relative resistance was supported by a three-trial study, where it was demonstrated that oral infection of 6 logs *L. monocytogenes* CFU/mL on day 14 or day 35 broiler chickens did not result in *L. monocytogenes* recovery in the gastrointestinal tract for two of the trials on day 42, while day 1 infections resulted in the recovery of the pathogen [63]. There are no characteristic signs of listeriosis in poultry despite occasional meningoencephalitis and symptoms from septicemia [61]. As such, the infection is generally subclinical and can pass on through the environment and onto the eggs without persistent symptoms throughout the flock [61,64]. Gastrointestinal infection results in contaminated fecal material, which can contaminate the feed, water, and pen environment [65].

Chemaly et al. [66] investigated the prevalence of *L. monocytogenes* in French layer flocks. Dust from the environments of 200 laying hen flocks was sampled, with 88 flocks reared in cages and 112 reared in floor pens. Of the caged hens, 7 of 139 dust samples were positive for *L. monocytogenes,* whereas only 6 of 206 dust samples from the floor-reared flocks were positive [66]. Aury et al. [46] also surveyed French layer hen flocks but focused only on caged flocks with more than 1000 hens. By sampling the fecal material, they were able to determine that of the 84 flocks, 25 (31%) tested positive for *Listeria*. These levels of prevalence are similar to the prevalence of *L. monocytogenes* found in the fecal material of broilers, including free-range flocks [5,46,66,67]. Schwaiger [68] obtained 800 cloacal swabs from 10 organic and 10 conventional caged laying hen flocks and found a *Listeria* prevalence of 1.3% in organic layer flocks and 1.6% in conventional flocks [68]. Conversely, Fenollar et al. [69] did not detect *L. monocytogenes* from shells or contents of eggs from conventionally raised, organic, or backyard flocks. Table 1 summarizes the information in this section.

One may assume that this prevalence rate is more relevant for vertical infection of the eggshell. Indeed, Jones et al. [70] and Farber et al. [71] found *Listeria* prevalence rates of 1.8% and 6% on the eggshell. Among 88 eggs collected from a conventional caged flock, only 1 sample tested positive for *Listeria,* while in 74 free-range flock samples, only 2 tested positive [70]. This sampling occurred across a single flock representing each management type, with four sampling events conducted throughout the year [70]. Farber et al. [71] investigated eggshells after washing in Ontario (7 facilities) and Quebec (10) processing facilities and found that only 3 out of 50 samples tested positive for *Listeria*. Schwaiger et al. [68] also collected 800 eggs from the farms; they performed cloacal swabs and by pooling and found no *L. monocytogenes* on the eggshell or in the contents (*n* = 80 pooled samples). While these rates may seem low, they could still pose a potential risk, as 92 billion eggs were sold in 2017.

Domestic fowl have been suggested as potential vehicles to transmit *Listeria* to human beings [21]. Risk factors for caged hens being infected with *L. monocytogenes* were detailed by Aury et al. [46]. They concluded that the presence of pets on the production sites, such as on many backyard farms, increased the risk of *L. monocytogenes* in layers flocks. Weber et al. [72] indicated that *L. monocytogenes* could exist in pets, and they can play a role as a vector. Layer feed can also play a role in the transmission of *L. monocytogenes,* and this pathogen has been detected in pelleted feed [73,74]. Caged layer hens had a decreased risk of acquiring *L. monocytogenes* if deep pit storage and conveyor belt system was used to dispense with fecal material (11.8% positive) as compared to deep pit alone (16.7%) or conveyor belt with dunghill storage (42.9%) [46]. The poultry red mite (PRM), *Dermanyssus gallinae*, is a ubiquitous parasite infesting egg-laying hens and can be a vector for pathogens [75]. Until recently, it was not proven that the PRM served as a vector for *L. monocytogenes* [76]. Sioutas et al. [75] captured mites and homogenized and cultured them for *L. monocytogenes*. For *Listeria,* DNA was extracted from bacteria grown in Tryptone Soya Yeast Extract Agar, and the *hly* gene was amplified by polymerase chain reaction (PCR) [75]. Microbiological cultures and PCR assays were positive for *L. monocytogenes,* confirming that PRM can carry this foodborne pathogen [75]. Table 2 summarizes the factors that increase the prevalence of *L. monocytogenes* in flocks.

Garcia et al. [78] evaluated the impact of the use of cattle manure on the presence of *Listeria* in the environment and eggs in free-range flocks. No statistically significant differences in levels of *Listeria* in the environment and eggs were found between those raised without manure (15.7% positive) and those raised with manure in the pasture (17.1%) [78]. The density of stocking has also been related to contamination with *Listeria*; layers were maintained at low (4.02 m^2^/bird) or high (2.01 m^2^/bird), and the eggs and the environment were assessed for the presence of *Listeria* [77]. *Listeria* was detected on the shells of 7.5% of the eggs and in the contents of 2.5% of the eggs in the low-density population, as opposed to 17.5% on the shells and 7.5% of the contents in the high-density group, respectively [77]. However, the grass in both paddocks had only 3.7% of samples positive, and no nesting boxes tested positive [77].

Crespo et al. [79] reported an unusual case of listeriosis in chickens in a backyard flock in Washington state. The flock included 20 egg-laying chickens approximately eight months old; over a five-month period, seven birds had died, and five birds exhibited depression, anorexia, and panting [79]. Two of the dead birds were presented for necropsy, and *L. monocytogenes* 4b was cultivated from multiple organs; the source of the infection was not discovered [79]. The family continued to consume eggs from the flock, and no illnesses were reported among them. Many species of birds are susceptible to infection by *L. monocytogenes*, although clinical disease in birds is rare. It is important to note the possibility of transmission of this pathogen to humans via direct contact or preparing and consuming chicken meat or eggs from contaminated flocks.

Studies that have investigated the presence of *Listeria* in farm fresh eggs have indicated limited prevalence within the egg [68,70]. However, if *Listeria* is present on the shell, it can easily contaminate the product when the egg is cracked open for consumption [80]. This is not surprising since bacteria, in general, can penetrate the shell membrane of a table egg [81]. When in the presence of a liquid, bacteria can be pulled into the egg through negative pressure [82,83]. This is especially prevalent during the contraction of the egg due to changes in temperature from when it was initially deposited by the hen (42 °C) to ambient (cooler) temperatures, as well as from the transfer of eggs into a refrigeration unit (7 °C) [82,83,84]. While there is a dearth of information on the penetration capabilities of *L. monocytogenes*, Padron [85] (1990) demonstrated that 59% of eggs could be penetrated by *Salmonella* when table eggs were placed in contaminated nest box shavings for 10 min. Gantois et al. [86] determined that shell thickness played no impact on limiting the effects of *Salmonella* invasion, although a developed cuticle was far more important. The cuticle is the most important obstacle against bacterial contamination of the inner egg [86,87]. However, Board and Halls [88] found that 3.5% of the eggs they examined did not possess a cuticle, and 8% had no cuticle on the apex or blunt end of the egg). The cuticle forms in the last 1.5 to 2 h of eggshell formation in utero [89]. Penetration of eggs with bacteria is directly dependent on the cuticle; eggs with poor cuticle deposition are more readily penetrated, while those with well-developed cuticles were never penetrated [90]. Nakazawa et al. [91] determined that *L. monocytogenes* can pass through membrane filters of 0.45 µm in 6 to 24 h and through 0.2 µm filters in 5 to 6 days. Since the pores in eggshells average between 1 and 10 µm [92], *Listeria* could easily penetrate the eggshell.

Once inside, bacteria encounter the egg albumen before reaching the naturally dense yolk [42]. With a pH of 9.5 and the presence of conalbumin (ovotransferrin), an iron-binding protein that causes free iron scarcity, albumenis unfavorable for microbial growth [42,93]. Flagellated bacteria have been shown to move through albumen and reach the yolk surface, allowing for rapid reproduction [82]. While no tests have been performed to determine whether *L. monocytogenes* interacts with the egg albumen, the presence of flagella, iron scavenging proteins (FrvA), and a high pH tolerance (3.0 to 9.5) suggests the bacterium could survive and reproduce within the egg yolk if it were present [94,95,96,97].

## 6. *Listeria’s* Presence in Egg-Processing Facilities

Egg-processing facilities offer numerous opportunities for potential cross-contamination and the introduction of *Listeria* onto the shell of eggs. There are two types of egg-processing facilities, in-line and off-line [98]. In-line processing occurs at the same location as egg production, with the eggs being delivered directly for processing by an automated conveyor system [98]. In the off-line processing model, eggs are produced at satellite farms and delivered to the processing facility via trucks [98]. Funk [99] conducted egg cooling studies and determined that when individual eggs with initial temperatures from 33 C to 39 °C were placed in a cooler (10 °C), the eggs cooled by approximately 11 °C per hour. When placed in the cooler in wire baskets, eggs in the center lost 4 °C per hour as opposed to eggs in metal buckets, which cooled at half the rate [99]. Shell eggs can be produced in either an off-line or on-line facility; in an off-line facility, the hens are housed in one location, and the eggs are transported to the processing facility as opposed to in-line, where the hen houses are directly connected to the processing facility. Anderson et al. [100] found that an egg from an off-line facility had an internal temperature at the packaging phase of between 24.4 °C and 26.7 °C, while the internal temperature of on-line-produced eggs was 26.7 °C to 34.4 °C at the packaging phase. During storage at 7.2° C, it would require as much as six days for the center egg in a 30-case pallet to cool to 7.2 °C [100,101].

In the U.S., eggs must be washed and sanitized before being sold as shell eggs or sent to a breaker for further processing [35]. USDA requires that shell eggs be washed in water that is at least 20 °F (11.1 °C) warmer than the warmest egg on the processing line but not less than 90 °F (32.2 °C), followed by a rinse with an approved sanitizer at the same or higher temperature as the wash water [35]. Jones et al. [102] investigated three washing schemes for shell eggs using two separate washers; the washers used (1) hot water in both washers (HH), (2) hot water then cold water (HC), or (3) cold water in both washers (CC). Predictably, the lowest egg temperatures were found in the CC eggs (21.25 °C for in-line processing and 17.25 °C for off-line processing) [102]. Three wash water samples were positive for *Listeria* in the off-line facility (1 HC, 2 CC), but no *Listeria* was detected in the eggs [102]. The authors concluded that a warm water wash followed by a cool water wash could decrease egg temperature while not compromising bacterial safety [102].

Park et al. [21] examined the effects of acidic electrolyzed water (EO) and chlorinated water to inactivate *Salmonella* Enteritidis and *Listeria monocytogenes* on shell eggs. They used EO at 16, 41, and 77 mg/L of total available chlorine compared to chlorinated water with 45 or 200 mg/L residual chlorine. *Listeria* counts declined proportionality to increasing concentrations of EO water as compared to deionized water with a maximum reduction of 3.7 log CFU per egg [21]. The most effective treatment was 200 mg/L chlorinated water, reducing the populations of *Listeria* and *Salmonella* by 4.89 and 3.83 log CFU per egg, respectively [21].

Manfreda et al. [103] developed a system using hot air to decontaminate shell eggs. The eggs were positioned on rolling cylinders with two hot air generators above and a cold air generator below. As a treatment, the eggs were given two 8-second bursts of 600 °C air while cold air (20 to 25 °C) was blown from below; the bursts of hot air were given 32 s apart [103]. *L. monocytogenes* was reduced by 1.2 logs CFU/eggshell immediately after treatment. Eggs were stored at 20 to 25 °C for 28 days; *L. monocytogenes* numbers were below the detection limit on both treated and untreated eggs at the end of this storage period [103]. Initially, the researchers speculated that this was due to penetration of the pathogen into the egg, but this was proven not to be true as there were no *L. monocytogenes* detected in the egg contents, leading them to hypothesize that this pathogen simply does not survive well on shell eggs under these storage conditions [103].

Laird et al. [104] observed that *L. monocytogenes* was able to survive commercial egg-washing procedures and could be present in water used to wash eggs. In artificial wash water contaminated with *L. monocytogenes,* they observed less than a 1 log reduction in viable cells recovered in the presence of alkaline detergent at 33 °C, although a 3 log decrease was observed at neutral pH without detergent [104]. In a survey of two commercial egg processors, they isolated *L. innocua* from environmental samples from both plants; they concluded that *L. monocytogenes* could potentially survive commercial egg washing and inhabit the environment of egg processors [104]. However, Milillo et al. [105] suggested that there are differences between *L. monocytogenes* and *L. innocua* stress responses.

## 7. *Listeria* and Retail Shell Eggs

After washing, eggs are subjected to “candling”, which involves inspecting the interior of the egg as it is rotated over a bright light to determine if there are any cracked or dirty shells and ensure that the yolk and albumen are in good shape and that there are no blood or meat spots visible [98]. Eggs destined to be sold as shell eggs undergo packaging. Eggs can be “loosely packed”, which means they are placed in a 20- or 30-egg flat to be sold primarily to restaurants [98]. For eggs sold directly to consumers, some cartons hold 12 or 18 eggs; these cartons are made attractive to appeal to the customer [106]. Eggs intended for further processing are sent to breakers where the liquid contents are separated from the shells to be sold as liquid whole eggs (LWE), liquid and dried whites, and liquid and dried yolks [98]. 

Once the eggs leave the processing facility, they are placed in refrigeration, where any *L. monocytogenes* contamination already present could continue to proliferate [107]. Guzmán-Gómez et al. [108] obtained shell eggs of five commercial brands from ten retail outlets and compared the results of standard culture techniques with those from nested PCR (n-PCR). Three of the eggshell samples tested positive by PCR for *L. monocytogenes,* while none were positive on culture, leading the authors to suggest that more sensitive methods, such as the n-PCR, should be used as a standard method rather than relying on the culture technique [108]. Better detection methods are critical since the presence of *L. monocytogenes* on or in eggs at retail could pose a significant risk to susceptible populations.

This risk is particularly evident if minimal cooking is involved. For example, retail eggs may subsequently be consumed raw by the consumer in the form of protein shakes, or they may be used in additional processing and consumed in the form of raw cookie dough [32]. However, even if the table eggs are purchased by the consumer and subsequently cooked, this may not be sufficient to kill the bacterium [80]. Brackett and Beuchat [80] added 10^2^ CFU/g and 10^5^ CFU/g *L. monocytogenes* to eggs, fried the eggs “sunnyside up”, and scrambled the eggs to an internal temperature of 70 to 73 °C. Only a 0.4 log decrease was observed when frying the eggs “sunny side up” while scrambling the eggs resulted in a 3 log reduction or, in the case of the low contamination event, brought populations below the limit of detection (1 CFU/g). These data suggest that *Listeria* can survive the cooking process despite the 3 log reduction observed by scrambling. Jamali et al. [109] compared egg products purchased from “street hawkers” with those purchased at a hypermarket in Malaysia. The hypermarket food courts were expected to be more hygienic than the street hawkers, although *L. monocytogenes* has been isolated from RTE foods at hypermarkets in Malaysia [110]. Samples of eggs and egg products were taken from both venues and compared; of 21 samples from the street vendors, 4 were positive for *Listeria,* while 2 out of 21 were positive from the hypermarket; thus, purchasing foods from street vendors and hypermarkets poses almost equal risk [109]. Erdoğrul [111] detected *L. monocytogenes* in the contents of 5 of 123 (4.06%) quail eggs sampled.

## 8. Recent Developments for Interventions for *Listeria* on Shell Eggs

In the U.S., the USDA requires that shell eggs must be sanitized by a rinse equivalent to 100 to 200 ppm chlorine or the equivalent [112]. However, participants in the National Organic Program certification (NOP) are required to remove all chlorine compounds on the surface of organic products with a potable water rinse, increasing costs and waste for the farmer [35]. However, in the EU, eggs are not washed, and the use of any disinfecting rinse causes the eggs to be downgraded [113]. The use of ultraviolet (UV) light to decontaminate the surface of eggs has become an area of interest; UV light is lethal to most microorganisms because of the damage to nucleic acids (DNA and RNA) [114]. Unfortunately, some microorganisms can repair this damage and become viable again; therefore, UV treatment must be administered to completely disrupt the nucleic acid [114,115]. The use of UV-C light for decontamination of food products is allowed in both the U.S. and the E.U., although Germany limits the use of UV-C to water, produce, and hard cheeses [116]. Holck et al. [116] inoculated intact eggs on the shell with a culture of *L. monocytogenes* in a growth medium and subsequently treated them with either UV-C light or pulsed UV light treatments. Reduction in *L. monocytogenes* was between 1.8 and 3.7 log, depending on the UV dose, with increasing the UV dose resulting in a minor increase in reduction; results were similar for the pulsed UV light treatment, although the pulsed UV treatments resulted in a greater decrease in *L. monocytogenes* survivors [116].

Electrostatic sprayers have also gained attention; the electrostatic sprayer works by using a positively charged surface to attract the negatively charged fluid droplets, which assures an even coating of the surface and minimizes exposure time [117]. Russell [118] used an electrostatic sprayer to spray electrolyzed oxidative (EO) water onto eggs artificially contaminated with *L. monocytogenes*. The treatment reduced *L. monocytogenes* below detection limits on 8 (53.3%), 13 (86.7%), 12 (80%), and 14 (93.3%) eggs of 15 tested in four separate replications [118]. More recently, Jiang et al. [119] compared conventional spraying to electrostatic spraying for the delivery of disinfectants to shell eggs. Eggs were inoculated with a two-strain overnight culture of *L. monocytogenes* and allowed to dry before being treated with peroxyacetic acid (PAA, 0.1%), lactic acid (LA, 5%), lactic acid and citric acid blend (LCA, 2.5%), sodium hypochlorite (SH, 50 ppm), or SaniDate-5.0 (SD (a mixture of 5.3% PAA and 23% H_2_O_2_); 0.25%) [119]. No significant differences were detected in the reduction in *L. monocytogenes* among antimicrobials, with all treatments significantly reducing counts to 4.50 to 5.11 log CFU per egg compared with 5.81 log CFU per egg for the unsprayed control [119]. Electrostatic spraying significantly increased the reduction in *L. monocytogenes* for treatments with PAA, SH, and SD as compared to conventional spraying but not for eggs treated with LA and LCA [119]. More studies are needed to evaluate the efficacy of antimicrobials when applied electrostatically for inactivating foodborne pathogens on egg products.

## 9. *Listeria* and Liquid Egg Products

Approximately 30% of eggs produced in the U.S. are sent to “breakers” to be processed into liquid or dried products, including liquid whole egg (LWE), liquid or dried yolks, and liquid or dried whites [120]. Many breakers are in-line, receiving eggs via a conveyor belt from the receiving area, which is connected directly to the hen houses, although some may be off-line, receiving eggs via truck from satellite farms [120]. Eggs move to a breaking machine where eggs are individually grasped and broken to let the liquid contents drain into a trough, which leads to a filter and then into a holding tank where the LWE is cooled as rapidly as possible to 4 °C; pasteurization and packaging are the final steps [120]. Breaker machines are also capable of separating the yolk from the white; the egg is broken over a cup, which retains the yolk while the white drains through to a trough, and the yolks are dumped in a separate area [120]. Subsequent treatments of liquid yolk and whites are the same as for LWE.

Leasor and Foegeding [121] collected raw LWE samples from egg processors in 11 states in the U.S. over eight months. When cultured for the presence of *Listeria* spp., 15 of the 42 samples (36%) were positive for *Listeria*; *L. innocua* was isolated from all the positive samples, while *L. monocytogenes* was isolated from two samples [121]. Rivoal et al. [122] used pulsed-field gel electrophoresis (PGFE) to determine if *L. monocytogenes* was present in raw or pasteurized LWE after storage at 2 °C for 2 days and at the end of shelf life. They detected *L. monocytogenes* in 25 of 144 raw samples, 4 in the pasteurized sample, and 2 of 144 that were at the end of shelf life. Calderón-Miranda et al. [20] inoculated LWE with *Listeria innocua* to determine the effects of pulsed electric fields (PEF) with or without nisin. Inactivation by PEF alone resulted in a 3.5 log reduction in *L. innocua*; when the LWE was treated with nisin (100 IU/mL), *L. innocua* was reduced by 5.5 log [20].

Liquid eggs can be combined from as many as 15 to 20 eggs into a one-liter product [20]. Pasteurization of liquid eggs can be used to improve product safety [123]. Li et al. [124] investigated two liquid egg products (A and B) with a water activity of 0.76 and 0.82 and a viscosity of 183 and 119 centipoise/s, respectively. After inoculation with 10^9^ log CFU/mL, egg samples were subjected to 64, 66, 68, and 70 °C. At 70 °C, a D value of 0.133 min for product A and 0.74 min for product B was determined. At 64 °C, a D value of 0.440 min for product A and 0.364 min for product B was determined. These values were higher than for *Salmonella* at all temperatures, with 70 °C D-values being 0.035 min (product A) and 0.048 min (product B). Additionally, Bartlett and Hawke [125] utilized *L. monocytogenes* strains Scott A and HAL957E1 (an egg isolate) to test pasteurization processes on the LWE. High-temperature short-time pasteurization (HTST) conditions require a temperature of 60 °C held for 3.5 min for LWE; this process produced 1.7 and 4.4 log reductions in LWE inoculated with 10^6^ CFU/mL of *L. monocytogenes* Scott A or HAL957E1, respectively. However, reductions of only 0.2 and 0.6 log CFU/mL were observed when the eggs also contained 10% NaCl. These data indicate that depending on the egg product, thermal inactivation may not be sufficient, and further mitigation strategies may be necessary.

## 10. *L. monocytogenes* in Ready-to-Eat (RTE) Egg Products

Most ready-to-eat (RTE) egg products, such as egg patties, omelets, and scrambled eggs, are fully cooked and reach temperatures of 85 °C (185 °F) before packaging and freezing [126]. Since *L. monocytogenes* is adequately inactivated by cooking to an internal temperature of 70 °C (158 °F) for 2 min, this treatment is adequate [127]. However, there is still the possibility of postprocess contamination with *L. monocytogenes,* plus the consumer preference for unfrozen products, which increases the food safety risk.

In 2012, one million eggs distributed to 34 states in the U.S. were recalled due to possible *Listeria* contamination, although no illnesses were reported [128]. Minnetonka, Minn.-based Michael Foods announced the recall of 10- and 25-pound pails of eggs in brine (15,000 pails) after a third-party testing service indicated the products might be contaminated [128,129]. The eggs were marked as having a 45-day shelf life if the bucket had not been opened, and the eggs ended up in a variety of products, from egg salad to green salads [128]. The details of an internal investigation were not revealed, although the company stated that they believed a repair project in the packaging room to be the culprit [128]. In 2019, an outbreak of listeriosis linked to boiled eggs occurred in 34 states [130]. Eight people from five states were ill, with five hospitalizations and one death [130]. Thus, it would be desirable to develop antimicrobials for RTE egg products that will be distributed and stored in refrigerated storage.

There are limited data in the literature on antimicrobials or antimicrobial treatments for *L. monocytogene*s in RTE egg products. Shrestha et al. [126] recently studied the effects of nisin alone or in combination with various organic acids for the control of *L. monocytogenes* in cooked egg products. Liquid whole eggs were obtained from an egg supplier, cooked in pouches, and subsequently inoculated with a five-strain cocktail of *L. monocytogenes* [126]. In the nisin-only treatment, the various formulations were treated with 6.25 ppm nisin at differing pH levels. In products with a pH of 6.29, *L. monocytogenes* had less than a 2 log/g growth for 4 weeks; however, products with pH values of 7.42 or 7.84, as well as the untreated control (pH 7.34), supported a 4 log growth by 4 weeks [126]. These researchers also evaluated the effectiveness of nisin in combination with organic acids; 6.25 ppm nisin was combined with an acetate-based antimicrobial used at 1.0% *w*/*w* in egg formulation (A1.0), propionate at 0.3% (P0.3), acetate–diacetate blend at 1.0% (AD1.0), acetate–diacetate blend at 0.6% (AD0.6), and lactate alone at 2.0% (L2.0; as a positive control) [126]*. L. monocytogenes* did not grow in formulations A1.0 and AD1.0, whereas L2.0 and P0.3 supported a 2 log growth by weeks 6 and 15, respectively, and AD0.6 supported less than a 1 log growth over 20 weeks at 4.4 °C.

## 11. Conclusions

While more attention has been placed on *Salmonella* and *Campylobacter*, *L. monocytogenes* remains an important foodborne contaminant that can be isolated from poultry eggs and layer hens. *L. monocytogenes* can be found in both broilers and laying hens. It is doubtless that environmental contamination plays a critical role in *Listeria* contamination, especially in egg-processing plants, but the question remains whether live production or egg processing should be of greater concern. Future studies and detailed risk assessments are necessary for how *Listeria* contaminates the eggs, whether that is from vertical transmission and/or environmental factors. It would also be important to identify the potential cross-contamination sites to determine the best application of sanitation. In addition, more research is needed into post-processing treatments for RTE egg products and dishes made with raw eggs.

## Figures and Tables

**Table 1 microorganisms-11-02164-t001:** Incidence of *Listeria monocytogenes* found in flocks of chickens.

Type of Flock	Rearing Condition	% Positive	Reference
Laying hens	Cages	5	[66]
Laying hens	Floor reared	2.9	[66]
Broilers	Conventional	28.2	[66]
Broilers	Free-range	36.7	[66]
Laying hens	Cages	30.9	[46]
Broilers	Conventional	31.7	[46]
Laying hens	Caged (conventional)	1.6	[68]
Laying hens	Caged (organic)	1.3	[68]

**Table 2 microorganisms-11-02164-t002:** Factors identified as increasing prevalence of *Listeria monocytogenes* on poultry farms.

Factor	Reference
Pets	[46,72]
Feed	[73,74]
Manure storage system	[46]
Poultry red mite	[75]
Density of stocking	[77]

## Data Availability

Not applicable.

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
