# Peer review of "Listeria Occurrence in Conventional and Alternative Egg Production Systems"

_microorganisms, 2023, doi:10.3390/microorganisms11092164_

Round 1
Reviewer 1 Report
The manuscript presented by Ricke et al. present an extensive review of the possibilities for Listeria contamination of egg production and processing systems. The analysis performed shows the important of the identification of this cross contaminations and the impact that can have in our health and food system.
The manuscript is well presented, with scientific rigor and important analysis, however due to the lack of recent data I suggest its publication after major revision.
Some comments are raised below to correct some mistake or improve the quality of the paper
Line 121 to 134 – this paragraph should be reduced, as the incidence of Salmonella is not relevant for the paper. I would say that the 2 last sentences are the most important. Also, I suggest removing completely the sentence 125 to 126
Line 141 to 146 – This sentence needs more recent references. Please provide information preset in the last reports of monitoring agencies.
Overall, the manuscript needs to have more updated references in order to evaluate the real risk nowadays for the food industry.
Conclusions:
I suggest that the reference to Campylobacter in the conclusion should be removed. This pathogen was not mentioned. The conclusions should be only focus in Listeria. Please provide more inside about your conclusion.
The english Language is understandable, good quality
Author Response
Thank you for your suggestions. We have done our best to address all of your concerns, here are our detailed answers to your comments.
Reviewer 1
Comments and Suggestions for Authors
The manuscript presented by Ricke et al. present an extensive review of the possibilities for Listeria contamination of egg production and processing systems. The analysis performed shows the important of the identification of this cross contaminations and the impact that can have in our health and food system.
The manuscript is well presented, with scientific rigor and important analysis, however due to the lack of recent data I suggest its publication after major revision.
These more recent references have been added
Bitter, A. Why organic eggs are suddenly cheaper than conventional ones. Available at: https://uk.news.yahoo.com/why-organic-eggs-suddenly-cheaper-182324110.html?guccounter=1&guce_referrer=aHR0cHM6Ly93d3cuZ29vZ2xlLmNvbS8&guce_referrer_sig=AQAAACaVG_ZMnM9qjbJuR0tGDj-siLSA_XXhgj9-rjnNSNWMz_vgc8YjdUFFA0YPvfGzc_LmvYw3Qdpb5ELGn9VTnjvaSp_UyCadJTvu4jJ4yQR2O5O4w2AmTv1xSVXglMilIX-r0cRzvV6UMZsdFloLF05PGsd8XVRMUTiaLFFdNdLF Accessed 16 Aug 2023.
United Egg Producers. (2022). Facts & Stats. Available at: https://unitedegg.com/facts-stats/ Accessed 14 June 2023.
Walmart. 2023. Animal welfare. Available at: https://corporate.walmart.com/esgreport/esg-issues/animal-welfare#resources Accessed 14 Aug 2023.
Zaheer, K. An updated review on chicken eggs: Production, consumption, management aspects and nutritional benefits to human health. Food and Nutrition Sciences, 2015. 6(13). 1208-1220
ChickenFans.com. 2023. Poultry Industry Statistics (2023): Meat & Egg Production. Available at: https://www.chickenfans.com/poultry-industry-statistics/#Global-Egg-Production Accessed 14 June 2023.
Sioutas, G., Petridou, E., Minoudi, S. et al. Isolation of Listeria monocytogenes from poultry red mite (Dermanyssus gallinae) infesting a backyard chicken farm in Greece. Sci Rep 13, 685 (2023). https://doi.org/10.1038/s41598-023-27862-3
Sparagano, O. & Giangaspero, A. Parasitism in egg production systems: The role of the red mite (Dermanyssus gallinae). In Improving the Safety and Quality of Eggs. Egg Products Egg Chemistry, Production and Consumption (Woodhead Publishing Limited, 2011).
Garcia, J.S., Anderson, K.E., Guard, J.Y., Gast, R.K., Jones, D.R. Impact of organic dairy cattle manure on environmental and egg microbiology of organic free-range laying hens, Journal of Applied Poultry Research, Volume 30, Issue 4, 2021, 100189, ISSN 1056-6171, https://doi.org/10.1016/j.japr.2021.100189.
Garcia, J.S., Anderson, K.E., Guard, J.Y., Gast, R.K., Jones, D.R. Impact of paddock area stocking density of free-range laying hens on egg and environmental microbiology, Journal of Applied Poultry Research, Volume 32, Issue 2, 2023, 100338, ISSN 1056-6171, https://doi.org/10.1016/j.japr.2023.100338.
Shrestha, S., Erdmann, J.J., Riemann, M., Kroeger, K., Juneja, V.K., Brown, T. Ready-to-eat egg products formulated with nisin and organic acids to control Listeria monocytogenes. J Food Prot. 2023 86(5):100081. doi: 10.1016/j.jfp.2023.100081.
Some comments are raised below to correct some mistake or improve the quality of the paper
Line 121 to 134 – this paragraph should be reduced, as the incidence of Salmonella is not relevant for the paper. I would say that the 2 last sentences are the most important. Also, I suggest removing completely the sentence 125 to 126
Most of the paragraph in question has been removed
Line 141 to 146 – This sentence needs more recent references. Please provide information preset in the last reports of monitoring agencies.
More recent references added
Overall, the manuscript needs to have more updated references in order to evaluate the real risk nowadays for the food industry.
Updated references provided
Conclusions:
I suggest that the reference to Campylobacter in the conclusion should be removed. This pathogen was not mentioned. The conclusions should be only focus in Listeria. Please provide more inside about your conclusion.
Reference to Campylobacter removed
Garcia, J.S., Anderson, K.E., Guard, J.Y., Gast, R.K., Jones, D.R. Impact of paddock area stocking density of free-range laying hens on egg and environmental microbiology, Journal of Applied Poultry Research, Volume 32, Issue 2, 2023, 100338, ISSN 1056-6171, https://doi.org/10.1016/j.japr.2023.100338.
Shrestha, S., Erdmann, J.J., Riemann, M., Kroeger, K., Juneja, V.K., Brown, T. Ready-to-eat egg products formulated with nisin and organic acids to control Listeria monocytogenes. J Food Prot. 2023 86(5):100081. doi: 10.1016/j.jfp.2023.100081.
Some comments are raised below to correct some mistake or improve the quality of the paper
Line 121 to 134 – this paragraph should be reduced, as the incidence of Salmonella is not relevant for the paper. I would say that the 2 last sentences are the most important. Also, I suggest removing completely the sentence 125 to 126
Most of the paragraph in question has been removed
Line 141 to 146 – This sentence needs more recent references. Please provide information preset in the last reports of monitoring agencies.
More recent references added
Overall, the manuscript needs to have more updated references in order to evaluate the real risk nowadays for the food industry.
Updated references provided
Conclusions:
I suggest that the reference to Campylobacter in the conclusion should be removed. This pathogen was not mentioned. The conclusions should be only focus in Listeria. Please provide more inside about your conclusion.
Reference to Campylobacter removed
Reviewer 2 Report
This a manuscript review for the article titled ''Opportunities for Listeria Occurrence in Conventional and Alternative Egg Production Systems''
Indeed, there are not too many articles on Listeria and the egg environment, so this work is timely. The following issues shown below should be addressed to help improve the manuscript.
Lines 45-46 ''L. monocytogenes cells were present'' The text appears to be in a bigger font. Check.
Lines 40-41 ...........''23,000 cases of listeriosis occur annually worldwide''. The reference with this information is about 10 years. Is there any recent figures?
Lines 267-272: Information in this section is outdated. API system is not reliable for identification of Listeria. Replace with data that show molecular identification.
The only table in this article has nothing on eggs. Another table should be added to show egg infestation by Listeria at a glance. The species that are more dominant with egg infestation should be highlighted.
Section 3 is all about Salmonella. Where is the Listeria link? When compared to Salmonella, what is the position of Listeria.
Overall actual literature on occurrence of Listeria in eggs in the last 3-5 years is missing. This requires a major improvement.
The quality of English Language is fine.
Author Response
Thank you for your comments and suggestions. We have done our best to address all of your concerns. See our detailed answers below
Reviewer 2
This a manuscript review for the article titled ''Opportunities for Listeria Occurrence in Conventional and Alternative Egg Production Systems''
Indeed, there are not too many articles on Listeria and the egg environment, so this work is timely. The following issues shown below should be addressed to help improve the manuscript.
Lines 45-46 ''L. monocytogenes cells were present'' The text appears to be in a bigger font. Check.
Have checked and corrected font throughout manuscript
Lines 40-41 ...........''23,000 cases of listeriosis occur annually worldwide''. The reference with this information is about 10 years. Is there any recent figures?
We have replaced with the following.
In 2021, the last year for which data is available, 2,268 confirmed cases of listeriosis were reported by 30 EU/EEA Member States, a rate of 0.51 per 100,000 population [15] (European Center for Disease Prevention and Control, 2022). In that same year in the U.S. there were approximately 1,029 laboratory-confirmed domestically acquired cases, a rate of 0.31 per 100,000 [16] (Healthy People 2030, 2021).
Lines 267-272: Information in this section is outdated. API system is not reliable for identification of Listeria. Replace with data that show molecular identification.
This reference has not been replaced, but reference to API has been eliminated. Culture is still the gold standard for isolation and identification of L. monocytogenes, but we have added the following information.
The poultry red mite (PRM), Dermanyssus gallinae, is a ubiquitous parasite infesting egg-laying hens, and can be a vector for pathogens [75] (Sioutas et al., 2023). Until recently it was not proven that the PRM served as a vector for L. monocytogenes [76] (Sparagano & Giangaspero, 2011). Sioutas et al. (2023) [75] captured mites, homogenized them, and cultured them for L. monocytogenes. For Listeria DNA was extracted from bacteria grown in Tryptone Soya Yeast Extract Agar and the hly gene was amplified by polymerase chain reaction (PCR) (Sioutas et al., 2023) [75].
Guzmán-Gómez et al. [108] (2013) obtained shell eggs of five commercial brands from ten retail outlets and compared the results of standard culture techniques with those from nested PCR (n-PCR). Three of the eggshell samples tested positive for L. monocytogenes while none were positive on culture, leading the authors to suggest that more sensitive methods such as the n-PCR should be used as a standard method rather than relying on the culture technique [108] (Guzmán-Gómez et al., 2013). Better detection methods are critical since the presence of L. monocytogenes on or in eggs at retail could pose a significant risk to susceptible populations.
Microbiological cultures and PCR assays were positive for L. monocytogenes, confirming that PRM can carry this food-borne pathogen [75] (Sioutas et al., 2023).
The only table in this article has nothing on eggs. Another table should be added to show egg infestation by Listeria at a glance. The species that are more dominant with egg infestation should be highlighted.
Table has been eliminated
Section 3 is all about Salmonella. Where is the Listeria link? When compared to Salmonella, what is the position of Listeria.
References to Salmonella have been removed
Overall actual literature on occurrence of Listeria in eggs in the last 3-5 years is missing. This requires a major improvement.
We have added the following more recent references
Bitter, A. Why organic eggs are suddenly cheaper than conventional ones. Available at: https://uk.news.yahoo.com/why-organic-eggs-suddenly-cheaper-182324110.html?guccounter=1&guce_referrer=aHR0cHM6Ly93d3cuZ29vZ2xlLmNvbS8&guce_referrer_sig=AQAAACaVG_ZMnM9qjbJuR0tGDj-siLSA_XXhgj9-rjnNSNWMz_vgc8YjdUFFA0YPvfGzc_LmvYw3Qdpb5ELGn9VTnjvaSp_UyCadJTvu4jJ4yQR2O5O4w2AmTv1xSVXglMilIX-r0cRzvV6UMZsdFloLF05PGsd8XVRMUTiaLFFdNdLF Accessed 16 Aug 2023.
United Egg Producers. (2022). Facts & Stats. Available at: https://unitedegg.com/facts-stats/ Accessed 14 June 2023.
Walmart. 2023. Animal welfare. Available at: https://corporate.walmart.com/esgreport/esg-issues/animal-welfare#resources Accessed 14 Aug 2023.
Zaheer, K. An updated review on chicken eggs: Production, consumption, management aspects and nutritional benefits to human health. Food and Nutrition Sciences, 2015. 6(13). 1208-1220
ChickenFans.com. 2023. Poultry Industry Statistics (2023): Meat & Egg Production. Available at: https://www.chickenfans.com/poultry-industry-statistics/#Global-Egg-Production Accessed 14 June 2023.
Sioutas, G., Petridou, E., Minoudi, S. et al. Isolation of Listeria monocytogenes from poultry red mite (Dermanyssus gallinae) infesting a backyard chicken farm in Greece. Sci Rep 13, 685 (2023). https://doi.org/10.1038/s41598-023-27862-3
Sparagano, O. & Giangaspero, A. Parasitism in egg production systems: The role of the red mite (Dermanyssus gallinae). In Improving the Safety and Quality of Eggs. Egg Products Egg Chemistry, Production and Consumption (Woodhead Publishing Limited, 2011).
Garcia, J.S., Anderson, K.E., Guard, J.Y., Gast, R.K., Jones, D.R. Impact of organic dairy cattle manure on environmental and egg microbiology of organic free-range laying hens, Journal of Applied Poultry Research, Volume 30, Issue 4, 2021, 100189, ISSN 1056-6171, https://doi.org/10.1016/j.japr.2021.100189.
Garcia, J.S., Anderson, K.E., Guard, J.Y., Gast, R.K., Jones, D.R. Impact of paddock area stocking density of free-range laying hens on egg and environmental microbiology, Journal of Applied Poultry Research, Volume 32, Issue 2, 2023, 100338, ISSN 1056-6171, https://doi.org/10.1016/j.japr.2023.100338.
Shrestha, S., Erdmann, J.J., Riemann, M., Kroeger, K., Juneja, V.K., Brown, T. Ready-to-eat egg products formulated with nisin and organic acids to control Listeria monocytogenes. J Food Prot. 2023 86(5):100081. doi: 10.1016/j.jfp.2023.100081.
Round 2
Reviewer 2 Report
The manuscript has improved, but tables and figures, which are key features of a review, are missing. The irrelevant table has been removed. As advised previously, a relevant table that summarises the occurrences of Listeria in egg production conventional and alternative systems should be included. Authors can list such occurrences in the last 10 years or moreif possible, and mindicate where it occurred with the relevant reference.
Author Response
The manuscript has improved, but tables and figures, which are key features of a review, are missing. The irrelevant table has been removed. As advised previously, a relevant table that summarises the occurrences of Listeria in egg production conventional and alternative systems should be included. Authors can list such occurrences in the last 10 years or more if possible, and mindicate where it occurred with the relevant reference.
Tables have been added to the manuscript